# Accuracy of emergency medical service telephone triage of need for an ambulance response in suspected COVID-19: an observational cohort study

Carl Marincowitz,[1] Tony Stone,[1] Madina Hasan,[1] Richard Campbell,[1] Peter A Bath,[1,2] Janette Turner ![ORCID] ,[1] Richard Pilbery,[3] Benjamin David Thomas ![ORCID] ,[4] Laura Sutton,[4] Fiona Bell,[3] Katie Biggs ![ORCID] ,[4] Frank Hopfgartner,[2] Suvodeep Mazumdar,[2] Jennifer Petrie,[4] Steve Goodacre ![ORCID] [1]

For numbered affiliations see end of article.

**Correspondence to**
Benjamin David Thomas;
b.d.thomas@sheffield.ac.uk

## ABSTRACT

**Objective** To assess accuracy of emergency medical service (EMS) telephone triage in identifying patients who need an EMS response and identify factors which affect triage accuracy.

**Design** Observational cohort study.

**Setting** Emergency telephone triage provided by Yorkshire Ambulance Service (YAS) National Health Service (NHS) Trust.

**Participants** 12 653 adults who contacted EMS telephone triage services provided by YAS between 2 April 2020 and 29 June 2020 assessed by COVID-19 telephone triage pathways were included.

**Outcome** Accuracy of call handler decision to dispatch an ambulance was assessed in terms of death or need for organ support at 30 days from first contact with the telephone triage service.

**Results** Callers contacting EMS dispatch services had an 11.1% (1405/12 653) risk of death or needing organ support. In total, 2000/12 653 (16%) of callers did not receive an emergency response and they had a 70/2000 (3.5%) risk of death or organ support. Ambulances were dispatched to 4230 callers (33.4%) who were not conveyed to hospital and did not deteriorate. Multivariable modelling found variables of older age (1 year increase, OR: 1.05, 95% CI: 1.04 to 1.05) and presence of pre-existing respiratory disease (OR: 1.35, 95% CI: 1.13 to 1.60) to be predictors of false positive triage.

**Conclusion** Telephone triage can reduce ambulance responses but, with low specificity. A small but significant proportion of patients who do not receive an initial emergency response deteriorated. Research to improve accuracy of EMS telephone triage is needed and, due to limitations of routinely collected data, this is likely to require prospective data collection.

## STRENGTHS AND LIMITATIONS OF THIS STUDY

⇒ Despite concerns regarding accuracy in identifying need for emergency treatment, this is one of the first evaluations of emergency medical service (EMS) telephone triage protocols for suspected COVID-19.
⇒ Use of ambulance data linked to nationally collected death registrations and routinely collected healthcare data provided robust outcome data for all included callers.
⇒ Use of routine data limited assessment of factors which affect triage accuracy to those which are routinely collected.
⇒ The evaluation in this study is of a single UK ambulance service's implementation of EMS telephone triage protocols for suspected COVID-19 cases.

## BACKGROUND

During the COVID-19 pandemic, emergency medical service (EMS) call volumes have been volatile in the UK, Europe and North America. Some ambulance services in the UK reported up to three times the expected number of EMS calls during the first and second waves of the pandemic, an increase also observed in other parts of Europe.[1–4] Other EMS providers in the UK and North America observed initial decreases in call volumes.[5 6] Surges in demand in the second wave of the pandemic led to some ambulance services in the UK declaring major incidents and warning of care being compromised by overwhelming demand.[7 8]

EMS call handlers in the UK are typically trained non-clinical staff who use either the Advanced Medical Priority Dispatch System (AMPDS) or NHS pathways to triage the EMS response to calls.[4] On 2 April 2020, six English NHS ambulance services using AMPDS introduced a specific protocol for callers with suspected COVID-19 using the pandemic Card 36.[9 10] Structured questions

(summarised in online supplemental material 1) were used to triage urgency of EMS response into three levels: Delta (highest priority requiring immediate response), Charlie (medium priority requiring ambulance attendance when able) and Alpha (no ambulance dispatched). The AMPDS Card 36 was implemented by UK ambulance services in different ways, with some symptoms and selected high risk-groups receiving a higher priority response than normal, with alterations to the card implemented in the following months.[4]

There is evidence that changes in EMS practice due to the pandemic and delays in the emergency assessment of patients with COVID-19 may have contributed to avoidable deaths in the North West of England.[11] Despite the need for EMS telephone triage to balance ambulance responses to patients requiring life-saving interventions against an anticipation of overwhelming demand during the pandemic, there has been no previous evaluation of the accuracy of the clinical risk-assessment performed for patients with suspected COVID-19.

Our study aimed to:
1. Assess how accurately EMS telephone triage identified those likely to suffer an adverse outcome (death or organ support) and subsequently require an emergency ambulance response.
2. Identify any factors that may have affected the accuracy of EMS telephone triage.

## METHODS
### Study design and setting
This observational cohort study used linked routinely collected EMS telephone call centre data from Yorkshire Ambulance Service (YAS) National Health Service (NHS) Trust to assess the accuracy of clinical triage of patients with suspected COVID-19.

Emergency services provided by YAS cover a region in the north of England of approximately 6000 square miles and with a population of 5.3 million. In 2020/2021, YAS received more than 1000 000 emergency (999) calls.

### Data sources and linkage
YAS provided a data set of all 999 calls, triaged using a Card 36 pandemic triage pathway for patients with suspected COVID-19, received between the 2April and 29 June 2020. The data set consisted of patient identifiers, demographic data, call details and the outcome of the call (including whether an ambulance was dispatched) extracted from routinely collected electronic call records.

Health and social care data relating to the population in England within the UK NHS are managed by NHS Digital. We provided patient identifiers to NHS Digital to trace patients in our cohort and supply additional individual-level demographic, comorbidity and outcome data. NHS Digital identified records in their collections belonging to patients in our cohort, and provided data on patient demographics, limited COVID-related general practice (GP) records, emergency department attendances, hospital inpatient admissions, critical care periods and death registrations from the UK Office for National Statistics.

YAS and NHS Digital removed records where patients indicated that they did not wish their data to be used for research purposes, via the NHS data opt-out service.[12] The study team also excluded patients who had opted out of any part of the wider Pandemic Respiratory Infection Emergency System Triage (PRIEST) study,[13] of which this evaluation forms a part, and those with inconsistent records (eg, multiple deaths recorded or death before latest activity). Patient identifiers across all data sets were replaced with a consistent pseudo-identifier to enable the identification and linkage of records belonging to the same patient across all data sets but without revealing any patient identifiers. Calls which originated from other healthcare services (ie, where a decision that ambulance dispatch was required had already been made) were excluded from the study population.

### Inclusion criteria
Our final cohort consisted of all adult (aged 16 years and over) callers at time of first (index) EMS 999 call between 2 April and 29 June 2020 assessed using the suspected COVID-19 Card 36 triage pathway, who were successfully traced by NHS Digital.

### Outcome
The primary outcome was death, renal, respiratory or cardiovascular organ support (identified from death registration and critical care data) at 30 days from index contact.

The secondary outcomes were hospital conveyance following ambulance dispatch (transfer to ED or inpatient setting) and inpatient admission (recoded inpatient dataset) 30 days from index contact.

The 30-day duration of outcomes was based on the duration used when deriving the PRIEST clinical severity score.[13]

### Patient characteristics
Consistent with methods used to estimate the Charlson comorbidity index from the available routine data, comorbidities were included if recorded within 12 months before the index EMS call.[14 15] In a similar way, only immunosuppressant drug prescriptions documented in GP records within 30 days before the index contact, contributed to the immunosuppression comorbidity variable. Pregnancy status was based on GP records recorded in the previous 9 months. Frailty in patients older than 65 years was derived from the latest recorded Clinical Frailty Scale (CFS) score (if recorded) in the electronic GP records prior to index attendance.[16] Patients under the age of 65 years were not given a CFS score since it is not validated in this age group. However, in multivariable analysis, patients under the age of 65 were assumed to have a functional level equivalent to a mild frailty category (CFS 1–3).

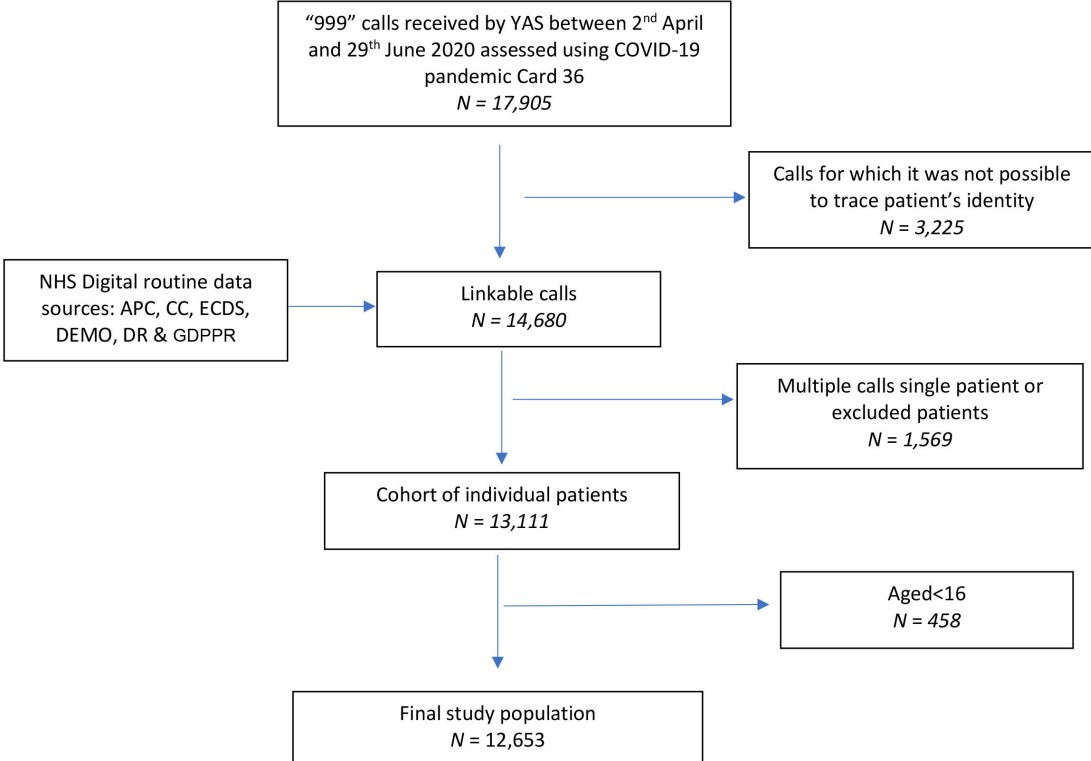

**Figure 1** STROBE flow diagram of selection of study population. APC, Admitted Patient Care; CC, Critical Care; DEMO, Demographics; DR, Death Registrations; ECDS, Emergency Care Data Set; GDPPR, General Practice Extraction Service Data for Pandemic Planning and Research; NHS, National Health Service; STROBE, Strengthening the Reporting of Observational Studies in Epidemiology; YAS, Yorkshire Ambulance Service.

## Analysis

We conducted a descriptive analysis of caller demographics, comorbidities and call disposition. The proportion of callers who experienced the primary and secondary outcome was estimated. To assess the accuracy of EMS telephone triage in identifying clinical outcomes requiring an emergency response, the call disposition of the index contact was divided into the binary classification of either: ambulance dispatched (Delta or Charlie priority); or other call outcome (Alpha priority) (online supplemental material 1). We assessed the accuracy of the binary triage classification (ambulance dispatched vs no ambulance dispatched) in terms of sensitivity, specificity, positive predictive value and negative predictive value (NPV) for the primary and secondary outcomes with 95% CIs.

Patient characteristics of false negatives (those who experienced the primary outcome (death or organ support) and no ambulance was dispatched) and true positives (ambulance dispatched who experienced the primary outcome) were compared. In patients with the adverse outcome (death or organ support) multivariable logistic regression was used to identify patient characteristics associated with false negative triage. We also compared the characteristics of false positives (ambulance dispatched and not conveyed to hospital) and true negatives (no ambulance dispatched) among those who did not experience the primary composite outcome. We

used multivariable logistic regression to identify factors which predicted false positive EMS triage. Obesity was excluded from multivariable analysis due to an observed implausible protective association with the primary outcome (death or organ support), which we believe to be an artefact of how these data were collected and recorded in the electronic GP data set.[17 18] Ethnicity and frailty were also excluded from multivariable analysis due to the high proportion of missing data. All analyses were completed using STATA V.16 (StataCorp. 2019. *Stata Statistical Software: Release 16*, College Station, TX: StataCorp LLC).

### Patient public involvement

The Sheffield Emergency Care Forum (SECF) is a public representative group interested in emergency care research.[19] Members of SECF advised on the development of the PRIEST study and two members joined the Study Steering Committee. Patients were not involved in the conduct of the study.

### RESULTS

All totals (including outcome) presented are rounded to the nearest 5, with small numbers suppressed to comply with NHS Digital data disclosure guidance.[20]

**Table 1** Population characteristics

| Population characteristic | Level | Whole population n=12 653 | No death or organ support (30 days)* n=11 250 | Death or organ support (30 days)* n=1405 |
|---|---|---|---|---|
| Age (years) | Median (IQR) mean | 66 (46–81) 62.3 | 63 (43–79) 60.6 | 80 (67–87) 70 |
| Sex (N, %)* | Male | 6260 (49.5%) | 5465 (48.6%) | 795 (56.5%) |
| Comorbidity (N, %)* | Cardiovascular disease | 810 (6.4%) | 690 (6.1%) | 120 (8.6%) |
| | Chronic resp. disease | 3730 (29.5%) | 3320 (29.5%) | 410 (29.4%) |
| | Diabetes | 2095 (16.5%) | 1770 (10.2%) | 325 (23.1%) |
| | Hypertension | 4545 (35.9%) | 6760 (15.7%) | 705 (50.4%) |
| | Immunosuppression (including steroid use) | 2540 (20.1%) | 3545 (31.5%) | 360 (25.8%) |
| | Active malignancy | 595 (4.7%) | 435 (3.9%) | 160 (11.5%) |
| | Obesity | 1960 (15.5%) | 1860 (16.5%) | 100 (7.3%) |
| | Pregnant | 135 (1.1%) | 135 (1.2%) | – |
| | Renal impairment | 365 (2.9%) | 310 (2.7%) | 60 (4.3%) |
| | Stroke | 285 (2.2%) | 240 (2.1%) | 45 (3.1%) |
| Social (N, %)* | Smoker | 4810 (38.0%) | 1740 (15.5%) | 550 (39.3%) |
| Number of prescribed drugs used* (N, %) | 0 | 3115 (24.6%) | 2940 (15.1) | 175 (12.4%) |
| | 1–5 | 6015 (47.5%) | 5320 (47.3%) | 700 (49.8%) |
| | 6–10 | 3030 (23.9%) | 2565 (22.8%) | 465 (33.1%) |
| | 11 or more | 490 (3.9%) | 425 (3.8%) | 65 (4.7%) |
| Clinical Frailty Scale (N, %)* | Unknown | 4560 (36.0%) | 3875 (34.4%) | 685 (48.9%) |
| | Aged<65 | 6105 (48.2%) | 5800 (51.5%) | 305 (21.6%) |
| | 1–3 | 165 (1.3%) | 150 (1.3%) | 15 (1.0%) |
| | 3–6 | 670 (5.3%) | 590 (5.2%) | 85 (6.0%) |
| | 6–9 | 1155 (9.1%) | 840 (7.5%) | 315 (22.5%) |
| Ethnicity (N, %)* | Asian or Asian British | 750 (5.9%) | 700 (6.2%) | 5 (3.6%) |
| | Black or Black British | 190 (1.5%) | 180 (1.6%) | 15 (0.9%) |
| | Mixed | 110 (0.9%) | 105 (0.9%) | – |
| | Other Ethnic Groups | 160 (1.3%) | 150 (1.3%) | – |
| | White | 9020 (71.3%) | 8070 (71.7%) | 950 (67.8%) |
| | Unknown | 2425 (19.2%) | 2050 (18.2%) | 375 (26.6%) |
| Deprivation Index (N, %)* | Unknown | 1250 (9.9%) | 1060 (9.4%) | 190 (13.5%) |
| | 1–2 | 4600 (40.3%) | 4155 (40.8%) | 440 (36.4%) |
| | 3–4 | 2075 (18.2%) | 1855 (18.2%) | 220 (18%) |
| | 5–6 | 1880 (16.5%) | 1660 (16.3%) | 220 (18%) |
| | 7–8 | 1695 (14.9%) | 1485 (14.6%) | 210 (17.2%) |
| | 9–10 | 1160 (10.2%) | 1030 (10.1%) | 130 (10.6%) |
| Ambulance dispatched (N, %)* | Ambulance | 10 650 (84.2%) | 9320 (82.8%) | 1335 (95%) |
| | No ambulance | 2000 (15.8%) | 1930 (17.2%) | 70 (5%) |
| Outcome (N, %)* | Death | 1155 (9.1%) | NA | 1155 (82.4%) |
| | Deaths due to COVID-19 (including after 30 days) | 690 (5.5%) | NA | 530 (37.8%) |
| | Organ support (within 30 days) | 335 (2.6%) | NA | 335 (23.8%) |
| Hospitalisation (N, %)* | ED attendance | 6945 (54.9%) | 5975 (53.1%) | 970 (69.1%) |
| | Inpatient admission | 5735 (45.3%) | 4650 (41.3%) | 1085 (77.3%) |

Continued

**Table 1** Continued

| Population characteristic | Level | Whole population n=12 653 | No death or organ support (30 days)* n=11 250 | Death or organ support (30 days)* n=1405 |
|---|---|---|---|---|
| Confirmed hospital diagnosis of COVID-19 (N, %)*† | In ED or as inpatient at 30 days | 1895 (15%) | 1300 (11.6%) | 595 (42.3%) |
| Time to primary outcome from index contact‡ up to and including (N, %) | 72 hours | 475 (3.8%) | NA | 475 (33.6%) |
| | 7 days | 780 (6.1%) | NA | 780 (55.5%) |

*To comply with NHS digital disclosure guidance totals for these variables are rounded to the nearest 5, which may result in apparent disparities in the overall totals.
†Unrestricted community testing for suspected COVID-19 was only available from 18 May 2020. Confirmed diagnosis is based on inpatient PCR testing or clinical diagnosis in hospital.
‡Suppressed due to small numbers.
ED, emergency department; NA, not applicable; NHS, National Health Service.

## Study population

Figure 1 and table 1 summarise study cohort derivation and the characteristics of the 12 653 included individual callers. In total, 1405 callers (11.1%, 95% CI: 10.5% to 11.7%) experienced the primary outcome (death or organ support) within 30 days following the index EMS 999 call. An ambulance was dispatched to 10 650 (84.2%) of callers. In our study cohort, 6070 patients (48%, 95%: 47.1% to 48.9%) were conveyed to hospital (ED or directly to an inpatient setting) and 5735 (45.3%, 95% CI: 44.5% to 46.2%) were admitted as hospital inpatients within 30 days of index contact.

The median age of the whole cohort was 66 (IQR=46–81), the cohort had an almost equal proportion of males (49.5%) and females (50.5%) and had high rates of comorbidity (chronic respiratory disease 29.5%, diabetes 16.6% and hypertension 35.9%).

## Accuracy of telephone triage of ambulance dispatch

Table 2 shows the accuracy of the EMS telephone triage decision to dispatch an ambulance for the composite primary outcome (death or organ support) indicating need for emergency intervention. Decision to dispatch an ambulance achieved a sensitivity of 95% (95% CI: 93.7% to 96.1%) to the primary outcome. If advised to self-care/non-urgent clinical assessment, the chance of experiencing an adverse outcome (death or organ support) was approximately 3.5% (NPV: 96.5%, 95% CI: 95.6% to 97.2%). The high sensitivity was achieved at the cost of specificity (17.2%–95% CI: 16.5% to 17.9%). Among patients for whom an ambulance was dispatched, the risk of serious adverse outcomes was 12.5% (95% CI: 11.9% to 13.2%), transfer to hospital was 57% (95% CI: 56% to 58%) and hospital admission 49.3% (48.3% to 50.2%).

## Prediction of false negative or false positive ambulance dispatch

Table 3 compares the characteristics of patients who experienced the primary outcome (death or organ support), and either did (true positives) or did not (false negatives) have an ambulance dispatched on index call. In both groups, over 50% of people experienced the primary adverse outcome within 7 days of first contact and around 33% of patients experienced the adverse outcome within 72 hours of index assessment. Multivariable modelling (online supplemental material 2) showed that female sex (OR: 1.89, 95% CI: 1.09 to 3.26) was associated with increased risk and, increasing age (OR: 0.95, 95% CI: 0.94 to 0.97) and malignancy (OR: 0.12, 95% CI: 0.02 to 0.92) reduced risk, of false negative triage.

**Table 2** Performance of decision to dispatch ambulance for composite primary outcome (death or organ support)

| Primary outcome 30 days (11.1%, 10.5%–11.7%) | | | |
|---|---|---|---|
| n=12 653 | Death or organ support | No death or organ support | |
| Ambulance dispatched | 1335 | 9320 | Sensitivity 95% (93.7%–96.1%) Positive predictive value 12.5% (11.9%–13.2%) |
| No ambulance dispatched | 70 | 1930 | Specificity 17.2% (16.5%–17.9%) Negative predictive value 96.5% (95.6%–97.2%) |

**Table 3** False negatives compared with true positives

| Population characteristic | Level | False negatives (no ambulance dispatch and primary outcome 30 days) n=70 | True positives (ambulance dispatch and primary outcome 30 days) n=1335 |
|---|---|---|---|
| Age (years) | Median (IQR) | 61.5 (51–80) | 80 (68–87) |
| | Mean | 64 | 76.7 |
| Sex (N, %) | Male | 35 (51.4%) | 760 (56.8%) |
| Comorbidity (N, %) | Cardiovascular disease | * | 115 (8.8%) |
| | Chronic resp. disease | 15 (20%) | 400 (29.9%) |
| | Diabetes | 20 (25.7%) | 305 (23%) |
| | Hypertension | 25 (34.3%) | 685 (51.2%) |
| | Immunosuppression (including steroid use) | 10 (17.1%) | 350 (26.3%) |
| | Active malignancy | * | 160 (11.9%) |
| | Obesity | 10 (12.9%) | 95 (7%) |
| | Pregnant | * | * |
| | Renal impairment | * | 55 (4.2%) |
| | Stroke | * | 40 (3.2%) |
| Social (N, %) | Smoker | 25 (32.9%) | 530 (39.6%) |
| Number of drugs used (N, %) | 0 | 20 (25.7%) | 155 (11.7%) |
| | 1–5 | 30 (42.9%) | 670 (50.2%) |
| | 6–10 | 20 (28.6%) | 445 (33.3%) |
| | 11 or more | * | 65 (4.8%) |
| Clinical Frailty Scale (N, %) | Unknown | 60 (84.3%) | 665 (49.8%) |
| | Aged<65 | * | 260 (19.7%) |
| | 1–3 | * | 15 (1%) |
| | 3–6 | * | 80 (6.2%) |
| | 6–9 | * | 310 (23.4%) |
| Ethnicity (N, %) | Asian or Asian British | * | 45 (3.5%) |
| | Black or Black British | * | 15 (1%) |
| | Mixed | * | * |
| | Other Ethnic Groups | * | 10 (0.7%) |
| | White | 45 (61.4%) | 910 (68.1%) |
| | Unknown | 20 (28.6%) | 355 (26.5%) |
| Deprivation Index (N, %) | Unknown | * | 185 (13.7%) |
| | 1–2 | 30 (46%) | 410 (35.8%) |
| | 3–4 | 10 (17.4%) | 205 (17.9%) |
| | 5–6 | * | 210 (18.4%) |
| | 7–8 | 10 (17.4%) | 200 (17.2%) |
| | 9–10 | * | 120 (10.6%) |
| Outcome (N, %) | Death | 45 (65.7%) | 1110 (83.3%) |
| | Deaths due to COVID-19 (including after 30 days) | 20 (27.2%) | 520 (39%) |
| | Organ support (within 30 days) | 35 (47.1%) | 300 (22.6%) |
| Hospitalisation (N, %) | ED attendance | 45 (65.7%) | 925 (69.2%) |
| | Inpatient admission | 50 (72.9%) | 1035 (77.6%) |

**Table 3** Continued

| Population characteristic | Level | False negatives (no ambulance dispatch and primary outcome 30 days) n=70 | True positives (ambulance dispatch and primary outcome 30 days) n=1335 |
|---|---|---|---|
| Confirmed hospital diagnosis of COVID-19 (N, %)† | In ED or as inpatient at 30 days | 25 (34.3%) | 570 (42.7%) |
| Time to primary outcome from index contact- up to and including (N, %) | 72 hours | 25 (37.1%) | 445 (33.5%) |
| | 7 days | 45 (62.8%) | 730 (54.9%) |

*Value suppressed due to small numbers.
†Unrestricted community testing for suspected COVID-19 infection was only available from 18 May 2020. Confirmed diagnosis is based on inpatient PCR testing or clinical diagnosis in hospital.
ED, emergency department.

Table 4 compares the characteristics of patients without the primary outcome (death or organ support) for whom an ambulance was dispatched and the patient was not transferred to hospital (false positives), or no ambulance was dispatched (true negatives); 33.4% of the cohort were false positives and online supplemental material 3 presents the results of multivariable modelling to identify factors associated with false positive triage. Risk of false positive was strongly associated with chronic respiratory disease (OR: 1.35, 95% CI: 1.13 to 1.60) and reduced in smokers (OR: 0.86, 95% CI: 0.75 to 0.99). Increasing age, deprivation and female sex were also associated with risk of false positive triage.

## DISCUSSION
### Summary
Our study showed that callers with suspected COVID-19 who made an EMS call had a high rate of adverse outcomes (death or organ support) (11.1%, 95% CI: 10.5% to 11.7%). This is around four times greater than the adverse outcome rate in callers who contacted the NHS 111 telephone service and half that seen in an ED population with suspected COVID-19.[18 21] Ambulances were dispatched to the majority of callers (84.2%) and the decision to dispatch an ambulance achieved a sensitivity of 95% (95% CI: 93.7% to 96.1%) for the primary outcome (death or organ support). Callers for whom an ambulance was not dispatched had a 3.5% or around 1/29 (NPP: 96.5%–95% CI: 95.6% to 97.2%) risk of the primary composite adverse outcome. This equated to 70 callers who died or required organ support up to 30 days from first call, not initially being dispatched an ambulance. Our evaluation cannot account for instances where ambulance dispatch was not clinically appropriate despite a high-risk of significant adverse outcomes (eg, transfer of care to community services for palliative care).

The cost of the high sensitivity achieved by dispatching ambulances to 84% of callers was a low specificity to the primary outcome (death or organ support) (17.2%–95% CI: 16.5% to 17.9%). In total, 33.4% of the cohort had an ambulance dispatched, and were not subsequently conveyed to hospital and nor did they experience the primary outcome. We used multivariable analysis to identify predictors of false negative and false positive triage. The findings need cautious interpretation, given the limited information available during telephone triage, but suggest that some comorbidities (such as chronic respiratory disease) and increasing age may be overestimated as predictors of adverse outcome.

### Strengths and limitations
Although specific EMS telephone triage protocols have been introduced for patients with suspected COVID-19, this appears to be the first evaluation of triage accuracy.[4] Our study used a large cohort of patients identified from routinely collected EMS records and linked this to nationally collected, patient-level healthcare data to provide robust clinical outcomes. We have assessed performance in a cohort of patients with suspected infection which, in the absence of accurate universally available rapid COVID-19 diagnostic tests, reflects the population who must be clinically triaged by urgent and emergency care services.

We have evaluated the performance EMS telephone triage for patients with suspected COVID-19 implemented by the YAS NHS Trust. Although use of the Card 36 protocol was recommended nationally, there was variation in implementation between different ambulance services.[4] Our study used data from the first wave of the pandemic and it was not until later waves that some ambulance services came under significant pressures due to increased demand.[1] Differences in EMS telephone triage demand and population characteristics of callers with suspected COVID-19 in later waves of the pandemic may affect the estimated accuracy of EMS decision to dispatch an ambulance. The implementation of a senior clinical support model within the call centre, and the influence of this secondary triage both of incoming calls and to support on-scene ambulance decision-making was not quantified in this study. It was not possible to identify within the data set which of the cohort were reviewed by a senior clinician at the initial call.

**Table 4** False positive compared with true negatives

| Population characteristic | Level | False positive (ambulance dispatched, not conveyed to hospital and no primary outcome 30 days) n=4230 | True negative (no ambulance dispatched and no primary outcome 30 days) n=1930 |
|---|---|---|---|
| Age (years) | Median (IQR) | 66 (46–81) | 37 (28–53) |
| | Mean | 62.9 | 41.6 |
| Sex (N, %) | Male | 1920 (45.4%) | 1000 (51.7%) |
| Comorbidity (N, %) | Cardiovascular disease | 235 (5.5%) | 30 (1.6%) |
| | Chronic resp. disease | 1280 (30.3%) | 380 (19.5%) |
| | Diabetes | 600 (14.2%) | 155 (7.8% |
| | Hypertension | 1480 (35%) | 255 (13.1%) |
| | Immunosuppression (including steroid use) | 825 (19.5%) | 180 (9.4%) |
| | Active malignancy | 140 (3.3%) | 20 (0.9%) |
| | Obesity | 700 (16.5%) | 260 (13.4%) |
| | Pregnant | 50 (1.2%) | 40 (2.2%) |
| | Renal impairment | 110 (2.6%) | 15 (0.8%) |
| | Stroke | 90 (2.1%) | 15 (0.7%) |
| Smoking status (N, %) | Smoker | 1570 (37.1%) | 640 (33.2%) |
| Number of drugs used (N, %) | 0 | 1040 (24.6%) | 955 (49.3%) |
| | 1–5 | 2075 (49%) | 805 (41.7%) |
| | 6–10 | 970 (22.9%) | 150 (7.6%) |
| | 11 or more | 150 (3.5%) | 25 (1.4%) |
| Clinical Frailty Scale (N, %) | Unknown | 1520 (35.9%) | 150 (7.6%) |
| | Aged<65 | 2030 (48%) | 1735 (89.8%) |
| | 1–3 | 55 (1.3%) | * |
| | 3–6 | 245 (5.7%) | 25 (1.2%) |
| | 6–9 | 385 (9.1%) | 25 (1.2%) |
| Ethnicity (N, %) | Asian or Asian British | 250 (5.9%) | 180 (9.2%) |
| | Black or Black British | 60 (1.4%) | 50 (2.5%) |
| | Mixed | 35 (0.9%) | 30 (1.4%) |
| | Other Ethnic Groups | 50 (1.2%) | 65 (3.3%) |
| | White | 3080 (72.7%) | 1210 (62.7%) |
| | Unknown | 760 (17.9%) | 405 (20.9%) |
| Deprivation Index (N, %) | Unknown | 420 (10%) | 130 (6.6%) |
| | 1–2 | 1470 (38.6%) | 925 (51.3%) |
| | 3–4 | 720 (18.9%) | 320 (17.7%) |
| | 5–6 | 655 (17.2%) | 245 (13.5%) |
| | 7–8 | 560 (14.7%) | 190 (10.5%) |
| | 9–10 | 405 (10.6%) | 125 (6.9%) |

*Value suppressed due to small numbers.

Our study used routinely collected data and consequently there were high rates of missing data for some variables, such as ethnicity and frailty. This prevented inclusion of these and non-routinely collected variables in analysis. We have assumed that if comorbidities or medication use was not recorded, they were not present. We also assumed in multivariable analysis, patients under the age of 65 had a functional level equivalent to a mild frailty category (CFS 1–3). The mechanism of how data are collected and recorded in the routine data sets used means that there may be bias in the classification of patients. We have previously identified this for the obesity variable where the estimated prevalence of obesity in our cohort is 15% (half that reported in the national health survey) and an implausible protective association with adverse outcomes was observed.[17 18] As weight is not

comprehensively and consistently measured by GPs, estimated statistical effects may reflect unknown characteristics associated with a measurement being taken, rather than the variable itself.

## Implications

EMS telephone triage of ambulance dispatch achieved a higher sensitivity for adverse outcomes (death or organ support) than triage methods used for patient acuity in the ED and NHS-111 COVID-19 assessment pathways.[22 23] Our cohort had a baseline risk of 11.1% of the primary outcome (death or organ support). The EMS telephone triage system selected those with a lower, 3.5% risk of the primary outcome (death or organ support), not to have an ambulance dispatched. Given the high baseline risk, an alternative would have been to dispatch an ambulance to every caller. This would have led to around 2000 more ambulance being dispatched within the time-period (an 18.8% increase). The acceptable risk of deterioration following telephone triage is subjective and significant variation in risk tolerance between clinicians and public representatives has been demonstrated.[24] When discharging patients from the ED with chest pain, an acceptable risk of a subsequent major adverse cardiovascular events has been found to be 1% or less for the majority of ED clinicians.[25] The risk of false negative triage (3.5%) in our cohort may therefore be unacceptably high for some clinicians. However, this is within the context of a face-to-face assessment with the availability of hospital investigations where greater accuracy may be expected.

Even using EMS telephone triage of ambulance dispatch, around a third of ambulances were dispatched to callers who were not subsequently conveyed to hospital and did not experience the primary outcome. In later waves of the pandemic, EMS providers in the UK experienced significant increases in demand with several ambulance services declaring major incidents.[8 26 27] The accuracy of triage observed in our cohort may be the best that realistically can be achieved given the limited information available with telephone triage. However, any measures which can increase accuracy of EMS telephone triage are greatly needed.

Our exploratory multivariable analysis indicates that older age and presence of pre-existing respiratory disease were associated with a higher rate of false positive triage among those who did not have an adverse event (death or organ support), and therefore may be overestimated in the triage process as predictors of adverse outcomes. Being aged 65 years or over is included as a specific high-risk factor in the Card 36 pandemic triage tool.[28] Use of a higher age threshold, or age in conjunction with variables related to performance status, could improve the specificity of EMS ambulance dispatch to significant adverse outcomes.[13] Vaccination against COVID-19 or previous infection may also act to reduce the risk of serious adverse outcomes in suspected infection and these factors are not included in Card 36 assessment.[29]

Limited information is available during EMS telephone triage determining whether ambulance dispatch is required. The Card 36 pandemic triage tool used for this decision-making in patients with suspected COVID-19 is consensus based. Empirical research assessing the predictive effect of all available variables is required if accuracy of triage, especially in terms of current over-triage and risk of false-negative triage, is to be improved. Given the limitations of the available routine data we have identified in this study, this is likely to require robust prospective data collection to empirically develop and validate more accurate triage tools. The use of trained, non-clinical call handlers for ambulance dispatch contrasts with other telephone triage services.[30] Other models of assessment could improve accuracy and also require evaluation. The current high levels of demand that ambulance services are experiencing mean that even small gains in accuracy could have a large positive effect on both safety and managing increasing demand.

## Conclusion

EMS telephone triage of need for ambulance dispatch in the first wave of the pandemic identified a lower risk population to whom ambulances were not dispatched. They constituted 16% of the cohort and around 2000 more ambulances would have been dispatched if these callers received an EMS response. However, patients to whom an ambulance was not dispatched had a clinically significant risk of death or requiring organ support. As the pandemic has developed into later waves, ambulance services in the UK have come under significant sustained pressure due to increased demand. Research to improve the accuracy of EMS telephone triage, especially in terms of safety and specificity, is needed. Due to the limitations of available routinely collected data this is likely to require robust prospective data collection.

**Author affiliations**
[1]Centre for Urgent and Emergency Care Research (CURE), Health Services Research School of Health and Related Research (ScHARR), University of Sheffield, Sheffield, UK
[2]Centre for Health Information Management Research (CHIMR) and Health Informatics Research Group, Information School, University of Sheffield, Sheffield, UK
[3]Yorkshire Ambulance Service NHS Trust, Wakefield, UK
[4]Clinical Trials Research Unit (CTRU), Health Services Research School of Health and Related Research (ScHARR), University of Sheffield, Sheffield, UK

**Contributors** The idea for the study was conceived by SG, JT, TS, FB, PAB and CM. Data processing and linkage was completed by TS and RC. The analyses were completed by CM and MH with specialist statistical advice from SG, JT, LS and PAB. CM, TS, MH, RC, PAB, JT, RP, BDT, LS, FB, KB, FH, SM, JP and SG contributed to interpretation of results, read and approved the final manuscript. CM is the guarantor of the manuscript.

**Funding** CM is a National Institute for Health Research (NIHR) Clinical Lecturer in Emergency Medicine (grant number not applicable/NA). The PRIEST study was funded by the UK National Institute for Health Research, Health Technology Assessment (HTA) programme (project reference 11/46/07). This publication presents independent research funded by the National Institute for Health Research and University of Sheffield. The views expressed are those of the author(s) and not necessarily those of the University of Sheffield, the NHS, the NIHR or the Department of Health and Social Care.

**Competing interests** None declared.

**Patient and public involvement** Patients and/or the public were involved in the design, or conduct, or reporting or dissemination plans of this research. Refer to the Methods section for further details.

**Patient consent for publication** Not required.

**Ethics approval** The North West-Haydock Research Ethics Committee gave a favourable opinion on the PAINTED study on 25 June 2012 (reference 12/NW/0303) and on the updated PRIEST study on 23 March 2020, including the analysis presented here. The Confidentiality Advisory Group of the NHS Health Research Authority granted approval to collect data without patient consent in line with Section 251 of the National Health Service Act 2006. Access to data collected by NHS Digital was recommended for approval by its Independent Group Advising on the Release of Data (IGARD) on 11 September 2021 having received additional recommendation for approval for access to GP records from the Profession Advisory Group (PAG) on 19 August 2021.

**Provenance and peer review** Not commissioned; externally peer reviewed.

**Data availability statement** Data may be obtained from a third party and are not publicly available. The data used for this study are subject to data sharing agreements with NHS Digital and Yorkshire Ambulance Service which prohibits further sharing of individual level data by the research team. The data sets used are obtainable from these organisations subject to necessary authorisations and approvals.

**ORCID iDs**
Janette Turner http://orcid.org/0000-0003-3884-7875
Benjamin David Thomas http://orcid.org/0000-0002-6659-6930
Katie Biggs http://orcid.org/0000-0003-4468-7417
Steve Goodacre http://orcid.org/0000-0003-0803-8444

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
