## [Reviewer comments · BMJ Open]

ARTICLE DETAILS

TITLE (PROVISIONAL)	Accuracy of emergency medical service telephone triage of need for an ambulance response in suspected COVID-19: An observational cohort study
AUTHORS	Marincowitz, Carl; Stone, Tony; Hasan, Madina; Campbell, Richard; Bath, Peter; Turner, Janette; Pilbery, Richard; Thomas, Benjamin; Sutton, Laura; Bell, Fiona; Biggs, Katie; Hopfgartner, Frank; Mazumdar, Suvodeep; Petrie, Jennifer; Goodacre, Steve

VERSION 1 – REVIEW

REVIEWER	Tankel, Jeremy NHS Salford Clinical Commissioning Group
REVIEW RETURNED	15-Dec-2021

GENERAL COMMENTS	1. If I were to just read the results section in the abstract then it would be unclear what the results meant. Perhaps line 19/20 needs to be re-worded to help make the later sentence more clear.2. Page 5 line 57/58 would it read better if the phrase “adverse outcome” was reworded as “died”. I think that would be clearer.3. Ethical approval: well done in overcoming the arduous hoops this must have presented.4. Table 1. I think it might be a positive addition to the paper if an addition column was added for Non adverse outcome. Therefore you would compare/easily see those who survived with those who did not. Depending on how it looks you may then want to suppress the whole population data.5. For me the 70 people who died as a result of non dispatch is a significant number. As a primary care physician that is 70 very distressed families who will feel the service let them down. For me this needs to be reflected in the discussion/conclusions/recommendations. For me it indicated that the triage tool partially failed.6. Page 9 table 3. May I suggest that dropping the phrases” False negatives (No ambulance dispatch and adverse outcome) compared to true positives (ambulance dispatch and adverse outcome)” into the appropriate column headings of the table may make it an easier read.7. Ditto for table 48. This comment is about the number of deaths. In table 2 you refer to 70 deaths. In table 3 false negative column I don't see the same 70 deaths. For me the maths does not add up. Perhaps it is me who is not following the table but I was confused by this.9. Page 12 line 26/27. I would not agree with the word good. Good was still 70,deaths.10. In the discussion and implications/ I think the line of argument could be slightly clearer. There was a 3.5% failure rate that resulted in death. The triage tool is flawed. Revision is needed.
--

	Perhaps previous vaccination status/covid infection state could be incorporated. Better senior review could help. That is how I read the results. One final suggestion the primary outcome was death. The use of the phrase "primary outcome" detracts from that. Perhaps the phrase should be swapped when used elsewhere. 11. I found this a very important study and I really look forward to seeing it published.
--	--

REVIEWER	Enomoto, Yuki University of Tsukuba
REVIEW RETURNED	14-Feb-2022

GENERAL COMMENTS	This study examines the results of triage for emergency callers in the United Kingdom. The results showed high sensitivity and low under-triage, but not high specificity. This study has a large patient size, and the patient characteristics and outcomes are linked, providing important epidemiological information. This study will provide basic information for future research. Minor revision: Page4, Line 42 The citation for the PRIEST study would not be appropriate.
--

REVIEWER	Masuda, Yoshio National University of Singapore, Medicine
REVIEW RETURNED	27-Feb-2022

GENERAL COMMENTS	Thank you for the opportunity to review this observational cohort study. The paper is an interesting read. It is very well-written and easy to follow. In the context of the current pandemic, I believe it provides useful information on telephone triaging accuracy in Yorkshire, which can be translated to key information for respective stakeholders around the world. This is especially important given that telephone triaging is the first-line response that the general public uses to seek help. I do have a few comments, which can be found below: General Clarifications:  1. Are the durations of primary and secondary outcomes (30 days from index contact) pre-set in the system, or referenced from specific guidelines? If the latter is so, please reference appropriately. 2. Adults are defined as ages 18 and above in England, however, the study defines adults as aged 16 years and over. Is there a reason why? 3. I believe the assumption that patients under age 65 years to have a functional level equivalent to a mild frailty category (CFS 1-3) is valid, however, it should be explicitly mentioned in your limitations. 4. What are outcomes termed as "adverse outcomes"? Please define. Abstract:  1. Take out 'an' from "Callers had an 11.1% of the primary outcome". 2. Change "Multivariable modelling found older age and presence of pre-existing respiratory disease were significant predictors of false positive triage" to "Multivariable modelling found variables of
--

	older age and presence of pre-existing respiratory disease to be significant predictors of false positive triage”. 3. I would remove the first sentence of the conclusion – it is just a repeat of the results. Add in the portion where the authors recognised that research is needed to improve the accuracy of EMS telephone triage, however, due to limitations of available routinely collected data, this is likely to require robust prospective data collection. 4. Under the last point of the strengths and limitations of this study, change to “The evaluation in this study is of a single United Kingdom (UK) ambulance service’s implementation of EMS telephone triage protocols for suspected COVID-19 cases.” Background: 1. A small point but define the first abbreviation of “UK” as “United Kingdom”. 2. The sentence “EMS call handlers in the UK...to triage what emergency care response is needed to calls” sounds funny. Please rephrase it. 3. Replace “...and so need an emergency ambulance response” to “and subsequently require an emergency ambulance response”. Methods: 1. Could I clarify to what extent does this study form a part of the PRIEST study? Are all patients included in this study from the PRIEST study or there is an inclusion of non-PRIEST study patients? 2. Superscript all dates. 3. Please cite the study that showed obesity to be a variable associated with observed implausible protection association with the primary outcome in the ‘Methods’ section. I believe the authors did so in the ‘Limitations’ section. Results: Good and comprehensive. Discussion: 1. Personally, I am interested in the predictors of false negative and false positive triage. Although limited information is available, are there any proposals that the authors can suggest to counter this? 2. Strength, limitations and implications are valid and comprehensive. 3. I would suggest removing results from the ‘Conclusion’. References and Supplementary Materials: Good.
--	---

VERSION 1 – AUTHOR RESPONSE

Reviewer: 1

Dr. Jeremy Tankel, NHS Salford Clinical Commissioning Group

Comments to the Author:

1. If I were to just read the results section in the abstract then it would be unclear what the results meant. Perhaps line 19/20 needs to be re-worded to help make the later sentence more clear.

Response: The results section of the abstract has now been amended. We hope the meaning is now clearer.

2. Page 5 line 57/58 would it read better if the phrase “adverse outcome” was reworded as “died”. I think that would be clearer.

Response: We have now qualified our use of primary outcome throughout to indicate this means death or required organ support.

3. Ethical approval: well done in overcoming the arduous hoops this must have presented.

Response: Thank you.

4. Table 1. I think it might be a positive addition to the paper if an addition column was added for Non adverse outcome. Therefore you would compare/easily see those who survived with those who did not. Depending on how it looks you may then want to suppress the whole population data.

Response: The table has been amended as suggested.

5. For me the 70 people who died as a result of non dispatch is a significant number. As a primary care physician that is 70 very distressed families who will feel the service let them down. For me this needs to be reflected in the discussion/conclusions/ recommendations. For me it indicated that the triage tool partially failed.

Response: The primary outcome of our study was death or requirement for organ support in an HDU/ICU setting up to 30 days from first 999 call. With all triage methods, the risk of misclassifying a patient who deteriorates must be weighed against the implications for the service of using higher sensitivity methods which reduce the risk of missing a patient but increase over triage. We have tried to present the risk of false negative of triage in our cohort neutrally, along with the implications of using higher sensitivity methods (such as dispatching an ambulance to all callers given the high baseline risk), to allow readers to decide if risk of false negative triage was acceptable in this cohort.

In response to this comment and later comments (9 & 10), we have revised the discussion section throughout to: 1) clarify the primary outcome as death or organ support 2) highlight that 70 callers were not initially dispatched an ambulance who died or required organ support 3) that the risk of false negative triage may be unacceptably high (although risk appetites vary between different patients and clinicians). In particular, the first and fourth paragraph of the implications section have been extensively revised.

6. Page 9 table 3. May I suggest that dropping the phrases” False negatives (No ambulance dispatch and adverse outcome) compared to true positives (ambulance dispatch and adverse outcome)” into the appropriate column headings of the table may make it an easier read.

Response: This has been changed as suggested.

7. Ditto for table 4

Response: This has been changed as suggested.

8. This comment is about the number of deaths. In table 2 you refer to 70 deaths. In table 3 false negative column I don't see the same 70 deaths. For me the maths does not add up. Perhaps it is me who is not following the table but I was confused by this.

Response: Apologies that this isn't clear. The 70 callers who died or required organ support without initially being dispatched an ambulance (false negative triage) are indicated by N=70, in the heading of column 3 of Table 3. This is the same 70 callers with the primary outcome where an ambulance was not dispatched on first contact indicated in table 2.

9. Page 12 line 26/27. I would not agree with the word good. Good was still 70,deaths.

Response: This sentence has been amended.

10. In the discussion and implications/ I think the line of argument could be slightly clearer. There was a 3.5% failure rate that resulted in death. The triage tool is flawed. Revision is needed. Perhaps previous vaccination status/covid infection state could be incorporated. Better senior review could help. That is how I read the results. One final suggestion the primary outcome was death. The use of the phrase "primary outcome' detracts from that. Perhaps the phrase should be swapped when used elsewhere.

Response: Please see our response to comment 5, where we outline the changes we have made to the discussion in response to this and previous comments.

11. I found this a very important study and I really look forward to seeing it published.

Response: Thank you.

Reviewer: 2

Dr. Yuki Enomoto, University of Tsukuba

Comments to the Author:

This study examines the results of triage for emergency callers in the United Kingdom. The results showed high sensitivity and low under-triage, but not high specificity.

This study has a large patient size, and the patient characteristics and outcomes are linked, providing important epidemiological information. This study will provide basic information for future research.

Minor revision:

Page4, Line 42

The citation for the PRIEST study would not be appropriate.

Response: Thank you, this has been corrected.

Reviewer: 3

Mr. Yoshio Masuda, National University of Singapore

Comments to the Author:

Thank you for the opportunity to review this observational cohort study. The paper is an interesting read. It is very well-written and easy to follow. In the context of the current pandemic, I believe it provides useful information on telephone triaging accuracy in Yorkshire, which can be translated to key information for respective stakeholders around the world. This is especially important given that telephone triaging is the first-line response that the general public uses to seek help. I do have a few comments, which can be found below:

General Clarifications:

1. Are the durations of primary and secondary outcomes (30 days from index contact) pre-set in the system, or referenced from specific guidelines? If the latter is so, please reference appropriately.

Response: The 30 days from index contact used for the primary and secondary outcome was based on the duration of outcome used to derive the PRIEST score. This has now been referenced.

2. Adults are defined as ages 18 and above in England, however, the study defines adults as aged 16 years and over. Is there a reason why?

Response: Although adults are defined as aged 18 and above in England. 16-18 years olds have a different status to children legally and are defined as being young people. Most adult clinical pathways are used to treat young people (aged 16-18) and they would generally be treated in adult, as opposed to paediatric, clinical settings.

3. I believe the assumption that patients under age 65 years to have a functional level equivalent to a mild frailty category (CFS 1-3) is valid, however, it should be explicitly mentioned in your limitations.

Response: The 4th sentences of the 3rd paragraph of the strengths and limitations section has been added to state this.

4. What are outcomes termed as “adverse outcomes”? Please define.

Response: Adverse outcomes has been defined to encompass death or organ support throughout.

Abstract:

1. Take out ‘an’ from “Callers had an 11.1% of the primary outcome”.

Response: This sentence has been amended.

2. Change “Multivariable modelling found older age and presence of pre-existing respiratory disease were significant predictors of false positive triage” to “Multivariable modelling found variables of older age and presence of pre-existing respiratory disease to be significant predictors of false positive triage”.

Response: This sentence has been changed.

3. I would remove the first sentence of the conclusion – it is just a repeat of the results. Add in the portion where the authors recognised that research is needed to improve the accuracy of EMS telephone triage, however, due to limitations of available routinely collected data, this is likely to require robust prospective data collection.

Response: The conclusion of the abstract has been amended as suggested.

4. Under the last point of the strengths and limitations of this study, change to “The evaluation in this study is of a single United Kingdom (UK) ambulance service’s implementation of EMS telephone triage protocols for suspected COVID-19 cases.”

Response: This has been changed.

Background:

1. A small point but define the first abbreviation of “UK” as “United Kingdom”.

Response: This has now been defined.

2. The sentence “EMS call handlers in the UK...to triage what emergency care response is needed to calls” sounds funny. Please rephrase it.

Response: This sentence has been amended.

3. Replace “...and so need an emergency ambulance response” to “and subsequently require an emergency ambulance response”.

Response: This has been changed.

Methods:

1. Could I clarify to what extent does this study form a part of the PRIEST study? Are all patients included in this study from the PRIEST study or there is an inclusion of non-PRIEST study patients?

Response: The PRIEST study has two parts. The first part involved recruiting an Emergency Department cohort of patients to derive a risk-stratification tool for patients with suspected COVID-19. The second part involved using routine datasets to evaluate the accuracy of pre-hospital triage methods for patients with suspected COVID-19. This study is a component of the second part of the PRIEST study and uses a cohort of patients identified using ambulance service electronic health care records independently to recruitment of patients to the Emergency Department PRIEST study.

2. Superscript all dates.

Response: This has been corrected throughout.

3. Please cite the study that showed obesity to be a variable associated with observed implausible protection association with the primary outcome in the ‘Methods’ section. I believe the authors did so in the ‘Limitations’ section.

Response: These references have been added.

Results:

Good and comprehensive.

Discussion:

1. Personally, I am interested in the predictors of false negative and false positive triage. Although limited information is available, are there any proposals that the authors can suggest to counter this?

Response: We have expanded the final paragraph of the implications section where we make suggestions for future research, including the need for prospective data collection due to the limitations of routinely collected data.

2. Strength, limitations and implications are valid and comprehensive.

Response: Thank you.

3. I would suggest removing results from the ‘Conclusion’.

Response: The conclusion has been amended as suggested.

References and Supplementary Materials:

Good.

VERSION 2 – REVIEW

REVIEWER	Masuda, Yoshio National University of Singapore, Medicine
REVIEW RETURNED	31-Mar-2022

GENERAL COMMENTS	Thank you for addressing my comments. I am satisfied with the current version of the manuscript and recommend it for publication. I believe that the amended version is an excellent and comprehensive read.
--

REVIEWER	Tankel, Jeremy NHS Salford Clinical Commissioning Group
REVIEW RETURNED	01-Apr-2022

GENERAL COMMENTS	One minor correction I think. Table 1. Top line. Column 3 plus column 4 does not equal the total of column 2. I think this is a really good paper. Thank you.
--

VERSION 2 – AUTHOR RESPONSE

Reviewer: 1

Dr. Jeremy Tankel, NHS Salford Clinical Commissioning Group

Comments to the Author:

One minor correction I think.

Table 1. Top line. Column 3 plus column 4 does not equal the total of column 2.

I think this is a really good paper.

Thank you.

Response: Thank you highlighting. To comply with NHS digital disclosure guidance totals for our outcome and other variables are rounded to the nearest 5, which may result in apparent disparities in the overall totals. We have now added an explanation at the bottom of table 1 to explain this.

Reviewer: 3

Mr. Yoshio Masuda, National University of Singapore

Comments to the Author:

Thank you for addressing my comments. I am satisfied with the current version of the manuscript and recommend it for publication. I believe that the amended version is an excellent and comprehensive read.

Response: Thank you.